# An Efficient Detection Approach for Unmanned Aerial Vehicle (UAV) Small Targets Based on Group Convolution

Jianghao Cheng [1], Yanyan Liu [1], Guoning Li [2,*], Jin Li [3], Jiantao Peng [4] and Jintao Hong [5]

1 Department of Electronics and Information Engineering, Changchun University Science and Technology, Changchun 130022, China; 2020100813@mails.cust.edu.cn (J.C.); liuyanyan@cust.edu.cn (Y.L.)
2 Changchun Institute of Optics, Fine Mechanics and Physics, Chinese Academy of Sciences, Changchun 130033, China
3 Departamento de Óptica, Facultad de Física, Universidad Complutense, 28040 Madrid, Spain; jinli02@ucm.es
4 Shanghai Institute of Satellite Engineering, China Aerospace Science and Technology Corporation, Shanghai 201109, China; sise509sast@126.com
5 Department of Engineering, University of Cambridge, Cambridge CB3 0FA, UK; jh2101@eng.cam.ac.uk
* Correspondence: liguoning@ciomp.ac.cn

**Abstract:** To solve the problem that small drones in the sky are easily confused with background objects and difficult to detect, according to the characteristics of irregular movement, small size, and changeable shape of drones, using a regional target recognition algorithm, the structure characteristics of Group Convolution (GC) in Resnext50 are absorbed. The optimized GC-faster-RCNN is obtained by improving the Fast-RCNN algorithm and the following methods are performed. First, a clustering method is used to analyze the dataset, and appropriate prior bounding box types are obtained. Second, the Resnext50 is used to replace the original feature extraction network, and the improved channel attention mechanism is integrated into its network output to enhance its feature map information. Then, we calculate its effective receptive field according to the Feature Pyramid Network (FPN) structure and redesign the prior bounding box of the corresponding size to construct a multi-scale detection network for small targets. Experiments show that the algorithm has a recognition accuracy of up to 94.8% under 1080 P image quality, and a recognition speed of 8FPS, which can effectively detect the positions of 1–5 small UAVs in a picture. This method provides an effective positioning detection for low-altitude UAVs.

**Keywords:** cluster analysis; small UAV; channel attention; effective receptive field; transfer learning





## 1. Introduction

With the development of science and technology, the drone industry is becoming more and more improved. Drones have been used in a variety of daily tasks, such as agricu'ltural spraying, multi-angle photography, civil surveys, etc. To ensure that the use of UAVs is within the legal scope, the control of UAVs needs more effective methods. In the past, the detection approaches for UAVs were mainly based on radar and infrared detection technologies. However, radar detection [1] has blind spots, and due to the small size of the UAV itself, radar cannot effectively detect the orientation as well. Moreover, because the UAV is mostly driven by a brushless motor, the heat generation is very small. Therefore, infrared detection (i.e., infrared radiation) cannot be effectively detected. Its detection task is very challenging. The rise of deep learning in the new era has provided a new direction for the UAV field.

There are many kinds of deep learning target detection algorithms. A two-stage algorithm represented by the Faster-CNN [2,3] series is superior in detection accuracy. The approaches represented by Yolo [4] one-stage algorithms [5–7] are far ahead in detection speeds. However, their research on low-altitude small targets such as UAVs is very limited, and related research is very scarce. The lack of datasets further limits the development of

drone identification. Considering there is currently no unified drone dataset, this paper uses the dataset used in the literature [8] and some data collected by ourselves. Existing datasets are the main training object. In the dataset, the background contains a variety of strong and weak lighting, multi-shaped clouds, roads, houses, fields, and other complex scenes.

To solve the above problem that the UAV is difficult to detect, this paper develops an optimization processing method. The proposed method combines the attention mechanism with Resnext50 of the group convolution network structure to improve the processing efficiency of the network, then determines the type of prior bounding box according to the dataset, calculates the required prior bounding box size through the number of layers of the network, and finally shortens the training time and improves the practicability of detection through migration learning.

## 2. Related Work

The main algorithms in the field of UAV object detection [9,10] are mainly divided into two categories: one-stage algorithms were represented by Yolo and SSD; two-stage algorithms were represented by RCNN [11–13]. Although the first category has demonstrated that it is very efficient and its accuracy has a good performance in most detection tasks, it only works well in large- and medium-sized target detection and still performs poorly in the detection of small objects. Therefore, the development of the two-stage algorithm is necessary for the small object detection [14,15] case.

The recognition task of machine learning is actually a further operation on the basis of classification. Therefore, the implementation process of these two tasks has many similarities. For example, in the early training process, some simple underlying features are both learned. Therefore, the classification task can be used first. After training on some huge datasets, sufficient basic feature weights are obtained, and then the training of the basic weights can be directly skipped in the recognition task.

At the same time, since most of the repetitive work in the early stage of transfer learning does not need to be undertaken again, the hardware equipment requirements for subsequent training are also reduced. In general, the emergence of transfer learning not only greatly reduces the training time of subsequent recognition tasks, but also improves the efficiency and accuracy of training. Therefore, transfer learning is becoming more and more popular in the field of target detection.

## 3. Building a Drone Dataset

The selected UAV dataset is published by the Naval Postgraduate School (NPS) and is publicly available. The selected UAV dataset can be downloaded on the UAV Dataset home page (https://engineering.purdue.edu/~bouman/UAV_Dataset/, accessed on 15 November 2021). The dataset contains 50 videos in HD resolution (1920 × 1080 and 1280 × 760) of these drones with a minimum size of 10 × 8, an average size of 16.2 × 11.6, and a maximum size of 65 × 21. These videos are captured at a frame rate of 1 frame per second, and a total of 2303 frames are obtained, and then the pictures containing the target are selected from them.

Since this article is based on transfer learning training, the required dataset is not too large. In total, 1000 pictures were selected from the frame-by-frame pictures. Each picture contains 1~5 UAV targets, covering cloud occlusion and various complex scenes such as multiple lighting cloud reflections, morning, noon, and night images, air-to-ground and ground-to-air cases, etc. Then, we symmetrically process the images in the preprocessing stage to increase the generalization ability of the dataset.

## 4. Improved Faster-RCNN Algorithm

The specific contents of this section include the following: Section 4.1 elaborates on how to determine the type of a priori frame through the dataset; the characteristics and power of Resnext50 backbone are explained in Section 4.2; Section 4.3 introduces how to

calculate the size of prior bounding box through FPN; finally, the advantages and functions of the attention mechanism in the text are explained in Section 4.4.

In this paper, the prototype of the Faster-RCNN algorithm is used as the main body for improvement. First, the original network is replaced with a Resnext50, and then a multi-scale network is constructed by using FPN for each layer of the feature network to adapt to the detection of small UAVs in the sky in different scenarios. At the same time, the attention mechanism is integrated into the tail of its feature output layer to enhance the feature information of the target. Then, an appropriate prior bounding box size is designed according to the number of network layers and the size of the convolution kernel of the FPN.

The Resnext50 network can adopt a Group Convolution (GC) structure. Multiple parameters can be trained in the same inference time, and the parameters can be shared between groups. Only one set of parameters can be used for the whole group. In the designs of the parameters, it is much simpler than Resnet50. Therefore, the algorithm in this paper is called the GC-Faster-RCNN algorithm. Its main detection process is shown in Figure 1.

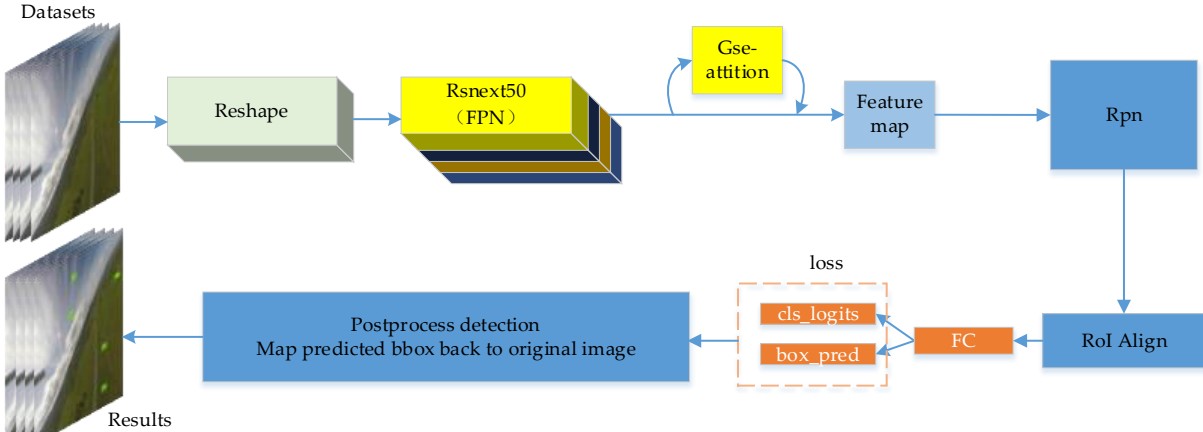

**Figure 1.** The detection flow chart of the improved Faster-RCNN algorithm.

*4.1. Types of Prior Bounding Boxes Using Cluster Analysis*

The cluster analysis method first needs to define the distance between samples. Common distances include absolute value distance, Euclidean distance, Minkowski distance, and Chebyshev distance. The Euclidean distance is one of the most common approaches.

Let the real box of the detected target be $box_1$ and the predicted box is $box_2$. The definition of the box value and the calculation formula of the IoU value are shown in the following:

$$box_1 = (w_1, h_1), \quad box_2 = (w_2, h_2) \tag{1}$$

$$IoU(b_1, b_2) = \frac{min(w_1, w_2) \cdot min(h_1, h_2)}{w_1 h_1 + w_2 h_2 - min(w_1, w_2) \cdot min(h_1, h_2)} \tag{2}$$

where $w$ and $h$ are the width and height of the center coordinates, respectively. $b_1$ and $b_2$ represent $box_1$ and $box_2$.

In the ordinary Kmeans algorithm, the Euclidean distance is used to complete the clustering. However, the judgment condition of the Euclidean distance is too single and absolute. The larger the object is, the larger the Euclidean distance is. We hope that the higher the degree of aggregation, the smaller the distance is. The Euclidean distance formula needs to be adjusted, as shown below:

$$Distance(b_1, b_2) = 1 - IoU(b_1, b_2) \tag{3}$$

The adjusted formula refers to the degree of coincidence between each cluster center and other boxes. If the original IoU $(b_1, b_2)$ is used, the larger the IoU $(b_1, b_2)$ is, the higher the degree of coincidence is. We hope that the higher the degree of coincidence is, the shorter the distance is. The Kmeans clustering method can be better. Therefore, the IoU $(b_1, b_2)$ cannot be used directly. The "1" should be added to the negative sign to ensure that the shorter the distance is, the better it is.

After the dataset in this paper was processed by the Kmeans elbow method, the results obtained are shown in Figure 2. It can be seen from Figure 2 that $K = 4$ is a relatively stable node. Therefore, the number of types of prior bounding boxes should be 4. The specific size of the prior bounding box is comprehensively determined by the number of convolutional layers of the FPN and the size of the receptive field.

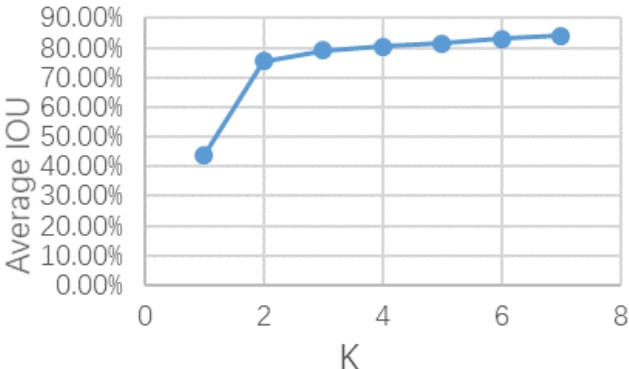

**Figure 2.** Kmeans clustering results.

*4.2. Resnext50 Backbone*

Deep learning has made great progress after the residual network structure was proposed. Its end-to-end characteristics basically exclude the limitation of the algorithm network depth. However, the excessive deep network structure will reduce the speed of algorithm training and prediction. In this paper, the Resnext50 network [16,17] is used. Due to its multiple parallel transmission structure, better accuracy can be obtained at the same network depth. Similar to the parallel processing of FPGA, it can solve the "queuing" phenomenon of data in the network. Moreover, its unique group convolution method enables parameter sharing between groups and reduces parameters. Therefore, the parallel group convolution structure of Resnext50 has higher efficiency at the same depth.

The network merges all groups at the output to ensure the integrity of the network. This paper makes appropriate adjustments to the parallel grouping of Resnet50 to ensure the optimization of the algorithm speed. Because the high-level network cannot provide a lot of useful information for the detection of small targets, in contrast, the low-level network can extract more effective information from small targets.

Let the width and height of the convolution kernel be $k$, the channel of the input matrix be $C_{in}$, $n$ is convolution kernels, and $g$ is the number of groups of group convolution. The principle of parameter optimization is as follows:

Parameters of Resnet50:

$$channel = n \qquad Parameters: k \times k \times C_{in} \times n \tag{4}$$

Parameters of Resnext50: (assuming three sets of branches):

$$channel = \frac{n}{g} \quad Parameters: \left( k \times k \times \frac{C_{in}}{g} \times \frac{n}{g} \right) \times g$$
$$= (k \times k \times C_{in} \times n) \times \frac{1}{g} \tag{5}$$

In contrast, the parameter is only one-fold of the original *g* (the more branches there are, the lower the parameter is), but the depth remains unchanged. Therefore, the detection speed is faster than the original. The specific data are shown in Table 1 below.

**Table 1.** Comparison of network data.

| Backbone | Layers | Parameters | Size (M) |
|----------|--------|-----------|----------|
| Resnet50 | 3 4 6 3 | 25,557,032 | 97.49 |
| Resnext50 | 3 4 6 3 | 25,028,904 | 95.48 |
| Resnext50 | 3 4 6 2 | 20,531,496 | 78.32 |

The specific network structure is shown in Figure 3. The Resnext50 network structure is a four-group mode. In Figure 4, *n* is the total number of convolution kernels, *g* is the number of groups of Resnext50, and $C_{in}$ is the number of input channels of the network.

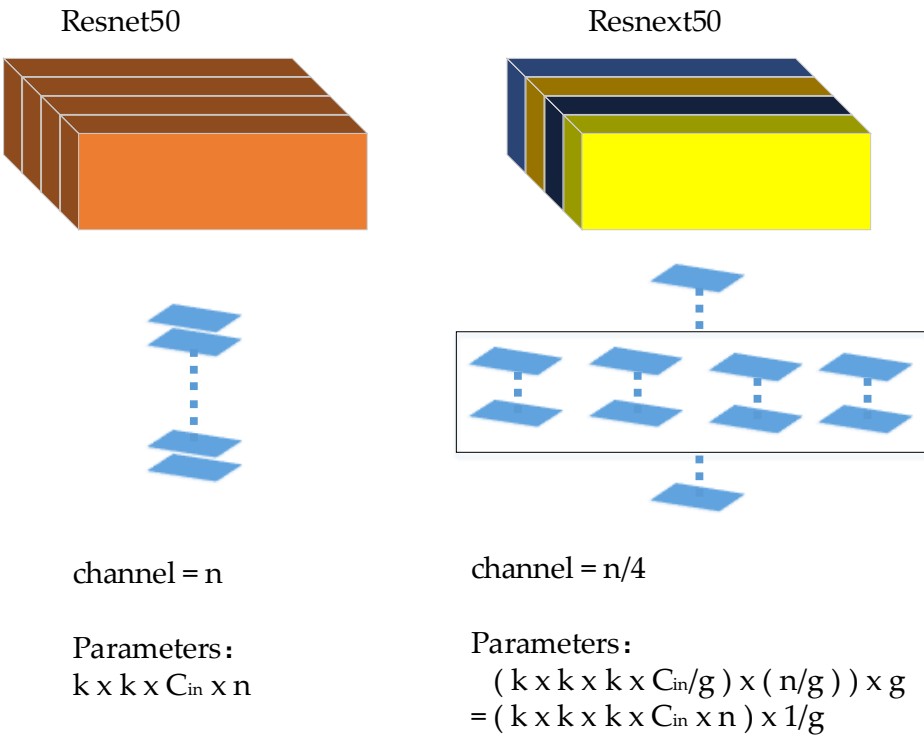

**Figure 3.** Resnet50 and Resnext50 structure diagram.

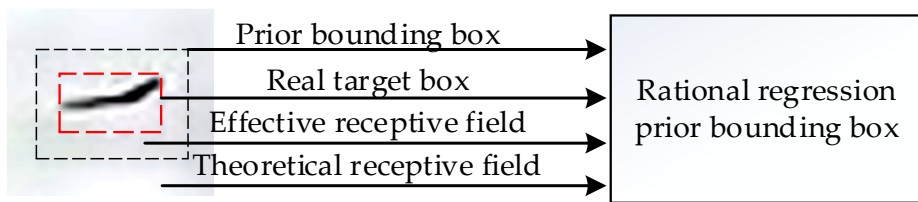

**Figure 4.** The relationship between prior bounding box and receptive field.

### 4.3. Prior Bounding Box Design for the Receptive Field Area in FPN

The receptive field is the size of the image input layer corresponding to an element in the feature map of the convolutional neural network. The size of the receptive field directly determines the range of information obtained by the feature network on the original image. The range of the theoretical receptive field is much larger than the effective receptive field.

As shown in Figure 4, the effective receptive field that plays a real role only accounts for a part of the theoretical receptive field.

The influence of the effective information in the receptive field on the output results is Gaussian distribution [18]. The information in the target detection center often occupies the peak position of the Gaussian distribution. Therefore, the design of the prior bounding box should be adapted to the effective receptive field. The size of the detection will directly affect the regression effect of the prior bounding box. A size too large or too small will cause the prior bounding box to deviate from the position of the original real frame, thereby reducing the detection precision.

In order to design a prior bounding box that is truly suitable for the effective receptive field, it is necessary to first calculate the size of the receptive field of the feature maps of each layer. The formula for calculating the receptive field is as follows:

$$
\begin{aligned}
&f(n) = (f(n+1) - 1) \times stride + ksize \\
&f(n)\ :\ N_{th}\ receptive\ filed \\
&stride\ :\ N_{th}\ layer\ step \\
&ksize\ :\ size\ of\ convolution\ kernel\ (pooled\ kernel)
\end{aligned}
\tag{6}
$$

According to the clustering results of Kmeans, this paper designed FPN [19] into four layers, and its structure is shown in Figure 5. Each layer corresponds to a type of prior bounding box. Here, the P2 layer of FPN is used as an example to illustrate how to reasonably design the size of the prior frame according to the calculated receptive field.

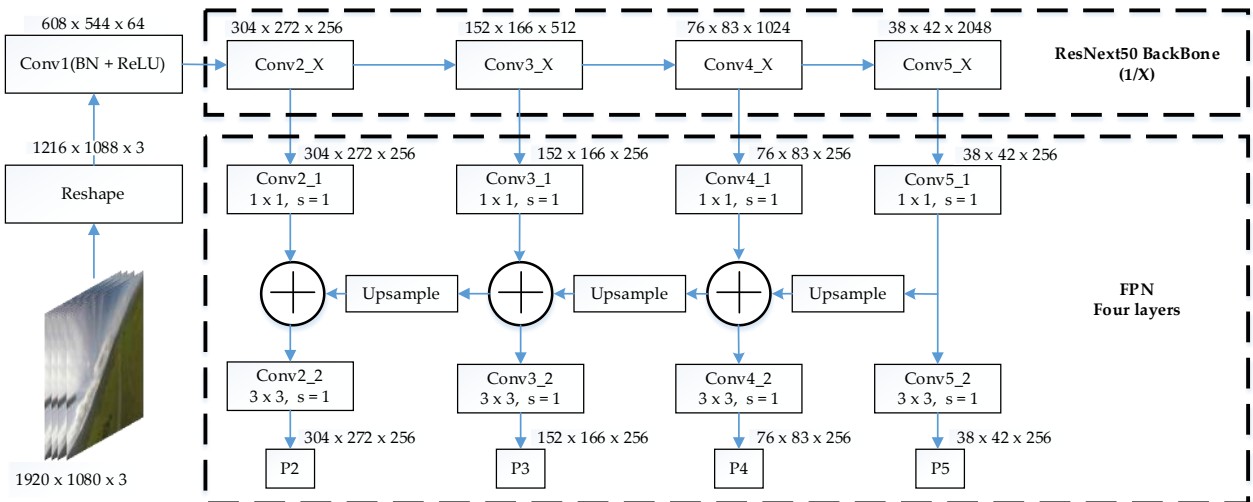

**Figure 5.** Resnext50 and FPN structure diagram.

P2~P5 in Figure 5 correspond to the predictions of the four prior bounding boxes, respectively. Since the second convolution layer of P2 uses a $3 \times 3$ convolution and a convolution kernel has a stride of 1, the node of the current prediction feature layer, the input area mapped to the previous layer has a size of $5 \times 5$, and then passes through a $1 \times 1$ convolution kernel with a stride of 1. The upper-level input is $3 \times 3$, which is located on the top layer of the feature network input. The size of the receptive field is $43 \times 43$. Therefore, P2's theoretical receptive field is $43 \times 43$. The receptive field of this layer is larger than the minimum UAV size and can be used to detect UAVs. P2 is the smallest receptive field. Therefore, effective receptive fields of the other three layers of FPN also meet the requirements. The average size of the UAV in this paper is $16^2$ or so, and the bottom layer of FPN has smaller target features. Thus, the minimum prior bounding box is designed to be the average size of the drone, to make full use of the low-level information to detect as many small drone targets as possible and calculate the size of all layers. After the receptive field, this paper set the prior bounding box size to four categories: $16^2$, $32^2$,

$64^2$, and $128^2$ to ensure that drones of each size can be selected. Since the overall size of the drone is between $10 \times 8$ and $65 \times 21$ during the period, the length–width ratio of the prior bounding box is designed with ratios of 2:1, 1:1, and 1:2 to ensure that the prior bounding box can match the appropriate target.

### 4.4. Gse Attention Mechanism

The attention mechanism [20] was originally a machine vision mechanism based on the bionics of the human eye. The aim is to quickly screen out effective information from a large amount of complicated information. It first appeared in the field of NLP and was later introduced into the field of CV. Its application method is different from other mechanisms. It is inserted in parallel to the original network and does not affect the original network structure. The insertion method is also very flexible and can be inserted into the head, middle, and tail of the network. The essence is to enhance the ability of the feature network to extract features by extracting weight information from different channels of the image and then integrating it into the output layer.

The specific process of the attention mechanism is to first perform feature compression through the global draw pooling method to obtain the global description features to overcome the disadvantage that the context information cannot be used because the receptive field of the convolution is too small. Then, the global features are activated to learn the relationship between each channel, and weights are extracted between different feature channels through the setting of nonlinear function parameters. Finally, the extracted feature weight result is multiplied by the output of the feature extraction network.

This paper uses an improved Gse-block, which is named after the initials of global pooling. The specific structure is shown in Figure 6. First, the input image is subjected to maximum pooling to retain the complete original features. Then, the low-level feature map of FPN is extracted. The weight, together with the weight extracted by the previous max pooling, is applied to the final output layer of the feature network. On the premise of ensuring the integrity of the feature, the feature enhancement of the shallow small target is strengthened. The detail features of the edge are improved, and the follow-up is reduced.

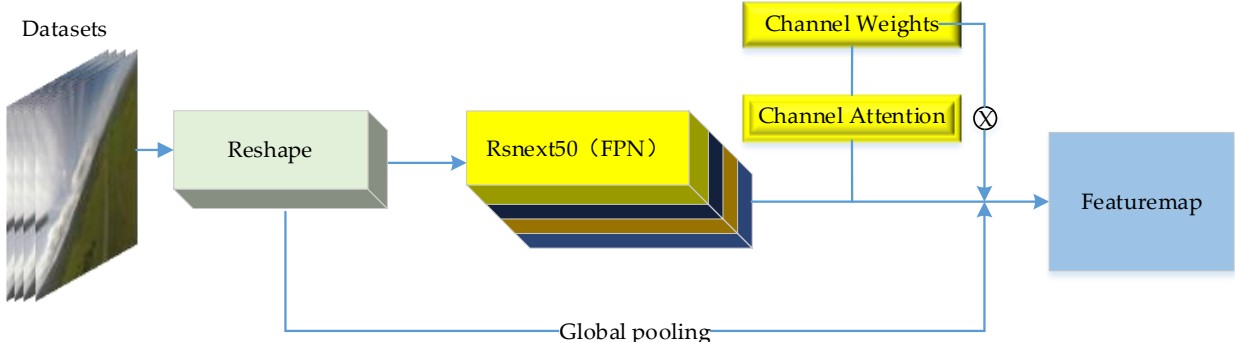

**Figure 6.** Gse-block structure diagram.

If the attention mechanism is directly added to the middle layer of the network, the structure of the network will be destroyed, and the pre-training operation cannot be performed. Therefore, this paper inserts the attention module into the tail of the feature network to ensure its integrity.

### 5. Experimental Results

The experimental platform configuration of this paper is i7-9700, Ubuntu18.04 system, the Python version used is 3.6.8, Pytorhc1.6.0 framework, and the training uses the 1660s Nvidia graphics card.

The dataset uses video software to intercept the original video file at one frame per second and select 1000 drone images from it. Among them, 750 images are used as the

training set and 200 images are used as the verification set. There are 50 images left as the test set.

### 5.1. Train the Network

Experiments were carried out on the algorithm structure before and after the improvement. The initial learning rate of 0.005 was used for training, and the training was performed according to the learning rate of 0.33 times in every 5 steps. In the accelerated stochastic gradient descent SGD method, momentum is added to suppress the oscillation of SGD, the initial value of the first-order momentum gradient is 0.9 to ensure that the accumulated descending direction is the descending direction of the current moment. Each batch contains 5 images. During training, random horizontal flip images are used to expand the dataset. At each time, the position information of the box is also synchronized.

### 5.2. Result Analysis

Since this experiment adopts the pre-training method, the training times of the algorithm before and after the improvement are 100~200 times, respectively, to achieve the fitting effect. The evaluation index system of a coco is used in this paper, and the specific training results are shown in Table 2.

**Table 2.** Comparison of detection accuracy of various network structures.

| Algorithm | Backbone (50 Layers) | mAP0.5/% | mAP0.75/% |
|---|---|---|---|
| Faster-rcnn | Resnet | 71.5 | 27.6 |
| | Resnet + Y | 89.8 | 38.2 |
| | Resnet + Y + Gse | 93.2 | 44.0 |
| GC-Faster-rcnn | Resnext + Y | 91.3 | 38.2 |
| | Resnext + Y + Gse | 94.8 | 41.9 |

Note: Y in the table is pre-training, Gse is the improved attention mechanism, GC refers to the algorithm after changing to the Resnex50 network.

It can be seen from Table 2 that the improved GC-Faster-RCNN in this paper has a greater improvement in accuracy compared with the original algorithm under the same conditions. The specific differences between the two methods are shown in Figure 7 below.

In Figure 7, the result on the left column is Faster-RCNN, and on the right is the GC-Faster-RCNN. It can be seen from the above figures that in the cloudy background, the UAV can be effectively detected, and the accuracy is higher than the Faster-RCNN.

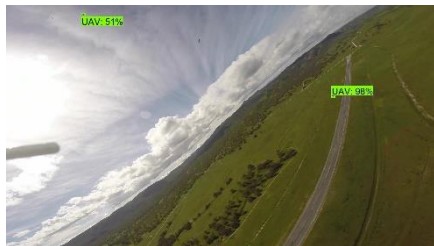
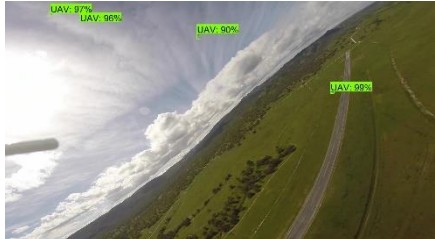

**Figure 7.** *Cont.*

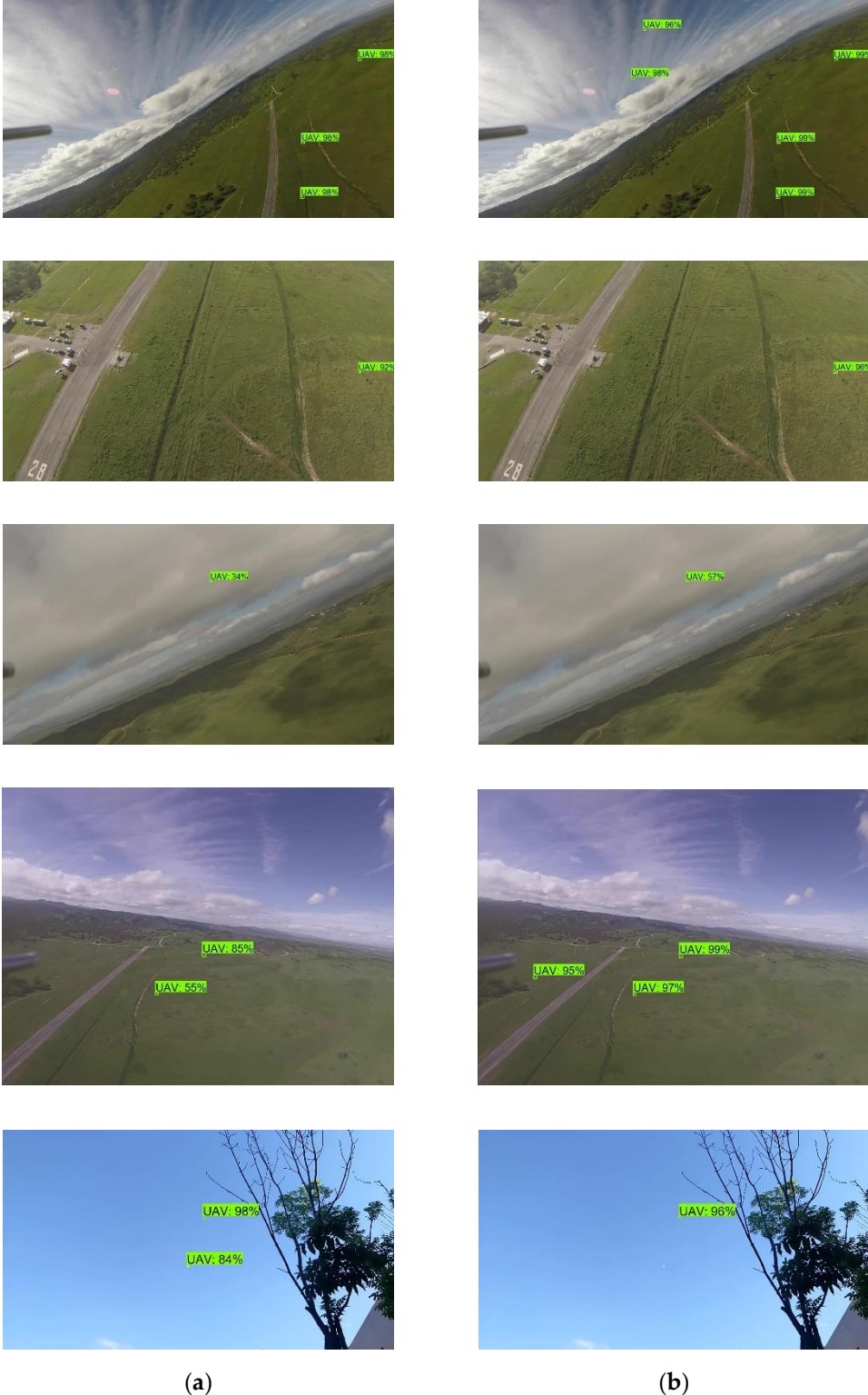

(**a**)                    (**b**)

**Figure 7.** Comparison chart of experimental results. (**a**) Faster-rcnn; (**b**) GC-Faster-rcnn. The boxes in these figures are the detection accuracy.

In the image taken by the mobile phone, because the color of the target and the background are very similar, and the target is smaller than the target in the dataset, and the leaves on the surrounding trees will also cause some detection interference to the detection, resulting in the coordinate information of the a priori frame potentially matching

other distractors during regression, so it is difficult to accurately predict the correct target. Therefore, the detection accuracy of the model for such scenes is reduced. The training loss and map curves in this paper are shown in Figure 8.

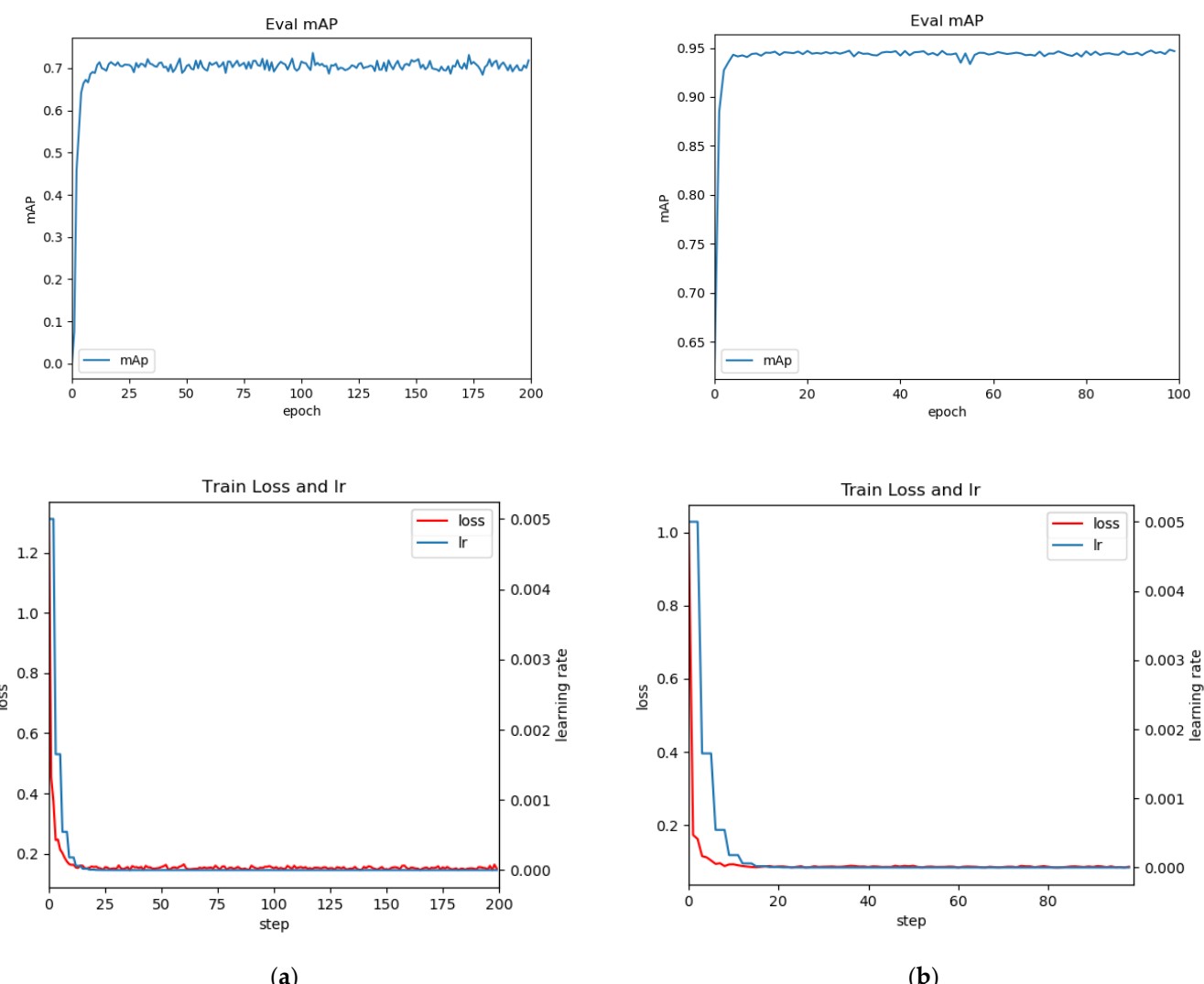

**Figure 8.** Comparison chart of training and loss curves. (**a**) Faster-rcn; (**b**) GC-Faster-rcnn.

According to the curve, the convergence of training and the decline in loss can be completed quickly with the blessing of pre-training, and the convergence can generally be completed in about 100 rounds.

It can be seen from Figure 8 that the improved GC-Faster-RCNN algorithm has a significant improvement on the map compared to the original algorithm, and the curve is more stable. Its accuracy can reach 94.8%.

Although the accuracy of fast RCNN can be stabilized in the 50th round, the subsequent overall accuracy still fluctuates greatly, and the detection objects of the original algorithm are all large, medium, and small targets. Therefore, the accuracy of small target detection is not as good as that of the algorithm only for small targets.

It is worth mentioning that the loss of the original algorithm decreases very fast, but it is still unstable until the training reaches 100 rounds. It reflects that the regression accuracy of its a priori frame fluctuates greatly and is not as stable as GC-Faster-RCNN.

At the same time, during the training, it is found that the simplification of the high-level network can also improve the accuracy of small targets, because the deep semantic information extracted by the high-level feature network is aimed at the characteristics of

medium and large targets, and the characteristic information of small targets has difficulty performing well in the high-level network after layer-by-layer network extraction. Therefore, the detection network of small targets can focus on the optimization of the low-level network, and the high-level can reduce it appropriately according to requirements.

## 6. Conclusions

This paper found, in the training process, that the detection of small targets is different from other targets and it needs targeted optimization. The bottom layer of the backbone feature extraction network is more suitable for the extraction of small target features because the information of small targets will be gradually lost in the progressive process of the network. Therefore, small targets are not suitable for network structures that are too deep. The a priori frame of the small target is also designed to be close to the size of the small target to ensure the accuracy of the regression. The attention mechanism is a parallel structure in the feature network, which can be inserted in the head and tail, but the enhancement effect in the head is not as good as that in the tail. The insertion process in the middle will destroy the network structure, and the pre-training cannot be used. If the pre-training is not used, it can be inserted in the middle. Therefore, the insertion process in the tail is a good choice.

The experimental results show that the detection accuracy of this paper is significantly improved. After using the pre-training method, the training speed of the model is very impressive, but the detection speed is slow, only 8FPS. In terms of the detection effect, the recognition effect of images with light interference from clouds in the air is better than that of images with many buildings on the ground, and many drone targets that are not easily detectable by the human eye can be directly detected.

There are still shortcomings in the detection in this paper: (1) ground buildings and vehicles interfere with the detection effect; (2) the detection speed still needs to be improved compared with the "Yolo series" of algorithms, and these shortcomings will be slowly addressed in the following research.

In the current trend of deep learning, many algorithms often need a large number of datasets to train models. The datasets required by some large models may reach the TB level, but they will also contain a lot of invalid information. The pre-training method is more efficient than self-training when dealing with small datasets and has a better effect when performing the same training task. Moreover, the pre-training method is also very popular in the current detection field. However, when there are a large number of labeled datasets, the effect of self-training is better than the pre-training, but it will also increase a lot of costs. Therefore, the selection of training methods should depend on the users' own applications. At the same time, cutting-edge experts are also very optimistic that high-quality small datasets can replace large datasets to train high-quality models in the future.

To the best of our knowledge, we believe that the pre-training approach can reduce the threshold of deep learning and allows more people to understand the technology of deep learning. Therefore, the pre-training approach still has good development prospects.

**Author Contributions:** Conceptualization, J.C., Y.L. and G.L.; methodology, J.C. and G.L.; software, J.C.; validation, J.C., Y.L. and G.L.; formal analysis, Y.L.; investigation, J.C. and G.L.; resources, J.P.; data curation, J.C.; writing—original draft preparation, J.C. and G.L; writing—review and editing, J.L. and J.H.; visualization, J.C.; supervision, Y.L.; project administration, G.L.; funding acquisition, Y.L. All authors have read and agreed to the published version of the manuscript.

**Funding:** Yanyan Liu would like to thank the Innovation Foundation of Changchun University of Science 358 and Technology (XJJLG-2018-07) and the Education Department of Jilin Province, China 359 (JJKH20210838KJ). Jin Li is grateful for financial support through a Marie Sklodowska Curie Fellowship, H2020-MSCA-IF-2020-101022219.

**Institutional Review Board Statement:** Not applicable.

**Informed Consent Statement:** Not applicable.

**Data Availability Statement:** Not applicable.

**Conflicts of Interest:** The authors declare that there are no conflict of interest to disclose.

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
