# Peer review of "An Efficient Detection Approach for Unmanned Aerial Vehicle (UAV) Small Targets Based on Group Convolution"

_applsci, doi:10.3390/app12115402_

Round 1

Reviewer 1 Report

Extensive editing of the English language is required. Therefore, reading this paper is really hard.

In this kind of paper, there is a classic question about the data set. Why this data set, and why only one? The link to download the data set is missing.

The authors also used self-collected images to evaluate the proposed algorithm. However, there is no reference to the image characteristics and the obtained results. 

The conclusions section (in this paper named Summarize) is confused. For example, what is mean: "The same tasks performed by data have great potential for development in the future, and cutting-edge machine vision experts are optimistic that high-quality small data can replace big data training."?

The references format is not according to the standards. 

Author Response

Reviewer 1

Q1: Extensive editing of the English language is required. Therefore, reading this paper is really hard.

Response: Thank you very much for your question. After reviewing the full-text sentence by sentence, many wrong words, wrong sentences, and where the sentence is not smooth are found. Now the full text has been modified sentence by sentence, and the text has been highlighted with yellow marks. Some unreasonable expressions have been completely corrected in the text, and try our best to achieve the correct expression of the meaning of the paper.

Q2: In this kind of paper, there is a classic question about the data set. Why this data set, and why only one? The link to download the data set is missing.

Response:

  1. At present, most of the data sets containing UAV small targets are 720p, so the image quality is not high. However, the image information contained in the 1080p data set in this paper is very clear in the details of the UAV target and background, the changes of illumination and background are presented in various time periods, and the clouds, background roads, houses, cars, and other small objects improve the anti-interference of UAV small target detection.

  1. Keeping the size of the data set consistent during the training of the target detection task can improve the speed and accuracy of the training and reduce the process of the feature network in preprocessing the pictures. Therefore, the dataset will use the same resolution pictures as much as possible. If the data sets with other different resolutions are mixed, the accuracy of the model may be affected.

  1. The data set used in this paper is published by the US naval Graduate School (NPS) and can be used publicly. Its link has been indicated in the revised manuscript. The link is shown on the right to view: UAV_ Dataset home page (purdue.edu)

Q3: The authors also used self-collected images to evaluate the proposed algorithm. However, there is no reference to the image characteristics and the obtained results. 

Response: We clarified the self-collected images for evaluating the proposed algorithm. In the self-collected data images, because they account for a relatively small proportion, at first it was just used to compare with the images in the data set, so the analysis part was ignored. Thank you for your advice, which makes us realize that this is wrong. So, in this paper, a certain analysis was made on the regression of the prior bounding box frame according to the size of the target and the interference factors in the image, such as the difference between the background color and target color, as well as the distractors trees and leaves in the scene. Then, according to the relationship between the training speed and the network feature layer in the training process, the adjustment of the feature layer is properly analyzed and summarized.

Q4: The conclusions section (in this paper named Summarize) is confused. For example, what is mean: "The same tasks performed by data have great potential for development in the future, and cutting-edge machine vision experts are optimistic that high-quality small data can replace big data training."?

Response: We revised the conclusion section in the revised manuscript. This means that the pre-training method is faster and more efficient than the self-training method under the same small data set training task. After consulting the relevant data, it is found that when there is a large enough and labeled data set, the self-training method is better than the pre-training and the results are more accurate, but it also brings high costs. Some experts put forward that, High-quality small data sets can make up for the gap between the size of data sets, so that small data sets with pre-training can also achieve the effect of using large data sets with self-training. However, high-quality small data sets also need manpower to screen and analyze, so the selection of large and small data sets still needs to be weighed by users. Here is our personal view on the pre-training method.

Q5: The references format is not according to the standards. 

Response: After checking the documents one by one, duplication and errors references are indeed found, so the duplicate documents have been deleted, the wrong documents have been changed again, and insert the references in the correct position.

The documents in the MDPI official template adopt the bibitem format, so the document format is to use overflow to convert the references in the bib format into the bibitem format on the MDPI official template

Reviewer 2 Report

Dear Authors

You have proposed a mechanism to solve the problem that small drones in the sky are easily confused with background objects and difficult to detect, due to the characteristics of the irregular movement, small size, and changeable shape of drones, using a regional target recognition algorithm. 

This research is so interesting, especially in these moments where the UAVs are used in several areas, including research, war, delivery, and more. 

The proposal is verified with real video datasets, obtaining a high percentage of precision. 

I suggest that improve the section o related works, with more information and make a comparison of these solutions. 

The last section can be written as conclusions, and not as a simple summary. 

Author Response

Reviewer 2

You have proposed a mechanism to solve the problem that small drones in the sky are easily confused with background objects and difficult to detect, due to the characteristics of the irregular movement, small size, and changeable shape of drones, using a regional target recognition algorithm.  This research is so interesting, especially in these moments where the UAVs are used in several areas, including research, war, delivery, and more. The proposal is verified with real video datasets, obtaining a high percentage of precision.  

Response: First of all, thank you very much for your positive suggestions. In recent years, with the development of technology, the UAV has entered people's vision more and more frequently. Moreover, due to its small size and fast start, the UAV is difficult to be effectively controlled, and it may also cause harm to the people below after failure in the air. At the same time, the UAV may also infringe on the privacy of others or countries. Therefore, the detection of the UAV as a small target is necessary for modern society, and we hope this paper can help people who want to know about small target detection.

 Q2: I suggest that improve the section o related works, with more information and make a comparison of these solutions.  

Response:  For your question, through the loss curve, we analyzed the regression accuracy of a priori frame and the number of characteristic network layers of the before and after the improvement algorithm. Because the fast RCNN algorithm is aimed at all detection targets of large, medium, and small size, there is no GC fast RCNN algorithm for small targets, which is more direct and accurate. It shows that the detection of small targets is different from other detection tasks, and targeted algorithms are needed to be competent. At the same time, it is found in the training that the small target is in the feature network. With the increase in the number of layers, the information will be less and less. The high-level network not only cannot provide sufficient small target information but also increases the burden of the network. Therefore, it is necessary to delete the high-level part of the feature network.

 Q3: The last section can be written as conclusions, and not as a simple summary.  

Response: For the last part, we improved the conclusions. We summarized some experience gained in training, such as the relationship between pre-training and data set, the ability of high-level and low-level for feature networks to extract target features, and the influence of attention in different positions on networks and features. At the end of this paper, we also analyzed the influence of the data set on training effects, how to select, and the prospect of future training methods.
